# Exploration of Compounds with 2-Phenylbenzo[*d*]oxazole Scaffold as Potential Skin-Lightening Agents through Inhibition of Melanin Biosynthesis and Tyrosinase Activity

**DOI:** 10.3390/molecules29174162

**Published:** 2024-09-02

**Authors:** Hee Jin Jung, Hyeon Seo Park, Hye Soo Park, Hye Jin Kim, Dahye Yoon, Yujin Park, Pusoon Chun, Hae Young Chung, Hyung Ryong Moon

**Affiliations:** 1Department of Manufacturing Pharmacy, College of Pharmacy and Research Institute for Drug Development, Pusan National University, Busan 46241, Republic of Korea; hjjung2046@pusan.ac.kr (H.J.J.); gustj6956@pusan.ac.kr (H.S.P.); hyesoo0713@pusan.ac.kr (H.S.P.); khj3358@pusan.ac.kr (H.J.K.);; 2Department of Medicinal Chemistry, New Drug Development Center, Daegu-Gyeongbuk Medical Innovation Foundation, Daegu 41061, Republic of Korea; pyj1016@kmedihub.re.kr; 3College of Pharmacy and Inje Institute of Pharmaceutical Sciences and Research, Inje University, Gimhae 50834, Republic of Korea; pusoon@inje.ac.kr; 4Department of Pharmacy, College of Pharmacy and Research Institute for Drug Development, Pusan National University, Busan 46241, Republic of Korea; hyjung@pusan.ac.kr

**Keywords:** tyrosinase, 2-phenylbenzoxazole, melanin, docking simulation, zebrafish larvae

## Abstract

Inspired by the potent tyrosinase inhibitory activity of phenolic compounds with a 2-phenylbenzo[*d*]thiazole scaffold, we explored phenolic compounds **1**–**15** with 2-phenylbenzo[*d*]oxazole, which is isosterically related to 2-phenylbenzo[*d*]thiazole, as novel tyrosinase inhibitors. Among these, compounds **3**, **8**, and **13**, featuring a resorcinol structure, exhibited significantly stronger mushroom tyrosinase inhibition than kojic acid, with compound **3** showing a nanomolar IC_50_ value of 0.51 μM. These results suggest that resorcinol plays an important role in tyrosinase inhibition. Kinetic studies using Lineweaver–Burk plots demonstrated the inhibition mechanisms of compounds **3**, **8**, and **13**, while docking simulation results indicated that the resorcinol structure contributed to tyrosinase binding through hydrophobic and hydrogen bonding interactions. Additionally, these compounds effectively inhibited tyrosinase activity and melanin production in B16F10 cells and inhibited B16F10 tyrosinase activity in situ in a concentration-dependent manner. As these compounds showed no cytotoxicity to epidermal cells, melanocytes, or keratinocytes, they are appropriate for skin applications. Compounds **8** and **13** demonstrated substantially higher depigmentation effects on zebrafish larvae than kojic acid, even at 800- and 400-times lower concentrations than kojic acid, respectively. These findings suggest that 2-phenylbenzo[*d*]oxazole is a promising candidate for tyrosinase inhibition.

## 1. Introduction

Melanin is a pigment that varies in color from a red-yellow to black-brown pigment and is found in nearly all living organisms, including bacteria, plants, animals, and humans [1,2]. Melanin is an oligomer or polymer. Eumelanin has a 5,6-dihydroxyindole structure, whereas pheomelanin contains benzothiazine and benzothiazole structures [3]. Melanin determines the color of the skin, eyes, and hair [4] and plays a protective role by absorbing ultraviolet rays [5]. However, excessive melanin pigmentation in certain areas can lead to cosmetic variations or medical problems such as freckles, melasma, senile spots, and hyperpigmentation [6,7]. Meanwhile, the production of melanin in fruits and vegetables acts as a representative factor that reduces quality via causing browning and unpleasant odors [8].

Melanin is synthesized through several chemical and enzymatic reactions [9]. Tyrosinase, tyrosinase-related protein-1 (TRP-1), and TRP-2 are involved in the melanin biosynthesis process, or melanogenesis [10]. Tyrosinase participates in the initial two oxidation processes using l-tyrosine and l-dopa as substrates during melanogenesis. TRP-1 and -2 are involved only in the synthesis of eumelanin. Among these enzymes, tyrosinase is reported to be a rate-determining enzyme in melanogenesis [11,12]. Thus, tyrosinase is an important target in the regulation of melanogenesis. Arbutin [13,14], kojic acid [15], hydroquinone [15], 4-butylresorcinol [13], tranexamic acid [13,16], and ascorbic acids [13,17] are representative tyrosinase inhibitors (Figure 1). Despite their use, the unmet demand for new whitening agents continues to rise because of their low clinical efficacy, instability during product manufacturing, and side effects [18,19,20,21].

Phenolic compounds with 2-phenylbenzo[*d*]thiazole scaffolds have been reported to display potent tyrosinase inhibitory activities against murine and mushroom tyrosinases (Figure 2) [22]. 2-Phenylbenzo[*d*]oxazole scaffolds are isosterically related to 2-phenylbenzo[*d*]thiazole. Additionally, compounds containing 2-phenylbenzo[*d*]oxazole exhibit various biological activities, including antimicrobial [23,24], antifungal [25], and anticancer [26] activities and the inhibition of transthyretin amyloid formation [27]. Therefore, phenolic compounds bearing a 2-phenylbenzo[*d*]oxazole scaffold were synthesized, and their abilities to inhibit tyrosinase activity, melanin production, and antioxidant activity were evaluated. In addition, their depigmentation effects in zebrafish larvae were examined. Furthermore, their inhibitory modes of action on tyrosinase and plausible chemical interactions with tyrosinase were examined.

## 2. Results and Discussion

### 2.1. Preparation of Target Compounds, 2-Phenylbenzo[d]oxazoles **1**–**15**

Through more than a decade of research on tyrosinase inhibitors, it has been established that 4-hydroxyphenyl, 2,4-dihydroxyphenyl, 3,4-dihydroxyphenyl, and 3-hydroxy-4-methoxyphenyl moieties play significant roles in inhibiting tyrosinase activity [28,29,30,31]. Consequently, for the synthesis of target compounds bearing the 2-phenylbenzoxazole scaffold, we introduced five 2-phenyl groups, including these phenyl moieties, into the target compounds. Three benzoxazoles, 6-methylbenzoxazole (series A), 6-chlorobenzoxazole (series B), and 5-methylbenzoxazole (series C), were introduced into the benzo[*d*]oxazole portion of the 2-phenylbenzoxazoles (Figure 1). Fifteen 2-phenylbenzoxazole compounds, **1**–**15,** were synthesized as previously reported [32]. Except for the target compounds with two hydroxyl groups on the 2-phenyl ring, the remaining target compounds were synthesized via two-step reactions. The condensation of 2-hydroxyanilines (2-hydroxy-4-methylaniline, 4-chloro-2-hydroxyaniline, or 2-hydroxy-5-methylaniline) with the appropriate benzaldehyde in ethanol produced the resulting imine intermediates. These intermediates were then equilibrated with 2,3-dihydrobenzo[*d*]oxazoles. Treatment with 2,3-dichloro-5,6-dicyanobenzoquinone (DDQ) oxidized the 2,3-dihydrobenzo[*d*]oxazoles to their corresponding benzo[*d*]oxazoles to afford the target compounds (**1**, **2**, **4**, **6**, **7**, **9**, **11**, **12**, and **14**). Further, the *O*-demethylation of the target compounds **2**, **7**, and **12** using BBr_3_ generated the target compounds **3**, **8**, and **13**, respectively, with a resorcinol group. The *O*-Demethylation of the target compounds **4**, **9**, and **14** using BBr_3_ afforded the target compounds **5**, **10**, and **15**, respectively, with a catechol group. The structures of the target compounds were confirmed by ^1^H and ^13^C NMR spectroscopy.

### 2.2. Mushroom Tyrosinase Inhibition of 2-Phenylbenzo[d]oxazole Compounds **1**–**15**

The inhibitory potency of compounds **1**–**15** against mushroom tyrosinase was examined using l-dopa and l-tyrosine as substrates. Overall, the IC_50_ value was lower when l-tyrosine was used as the substrate compared to when l-dopa was used as the substrate. The numbering of the phenyl ring at position 2 of benzoxazole followed the numbering of the common phenyl ring.

When l-tyrosine was used as a substrate, kojic acid exhibited an IC_50_ value of 14.33 ± 1.63 μM, indicating a strong inhibitory potency (Table 1). Among the 6-methyl-2-phenylbenzoxazole series of compounds (**1**–**5**), compound **1**, which had a hydroxyl group substituted at position 4 of the phenyl ring, showed a somewhat weak mushroom tyrosinase inhibition effect (IC_50_ value = 152.51 ± 14.33 μM). Introducing an additional hydroxyl group into the phenyl ring of **1** resulted in different effects, depending on the position of the substitution: compound **3** with a 2,4-dihydroxyphenyl ring showed a very high tyrosinase inhibition potency with a nanomolar IC_50_ value of 0.51 ± 0.00 μM, while compound **5** bearing a 3,4-dihydroxyphenyl ring exhibited a tyrosinase inhibition potency with an IC_50_ value of 144.06 ± 3.10 μM, similar to **1**. Interestingly, compound **2**, in which the 2,4-dihydroxyl group of the phenyl ring of **3** was substituted with a 2,4-dimethoxyl group, showed a very weak inhibitory potency (IC_50_ value > 200 μM). Replacing the 4-hydroxyl group on the phenyl ring of **5** with a 4-methoxyl group significantly reduced tyrosinase inhibition (IC_50_ value of **4** > 200 μM). The 6-Chloro-2-phenylbenzoxazole series compounds (**6**–**10**) and the 2-phenyl-5-methylbenzoxazole series compounds (**11**–**15**) showed tyrosinase inhibition trends similar to the 6-methyl-2-phenylbenzoxazole series compounds (**1**–**5**). Notably, compounds **3**, **8**, and **13,** each bearing a 2,4-dihydroxyphenyl ring, demonstrated the most potent mushroom tyrosinase inhibitory activities, with IC_50_ values of 0.51 ± 0.00, 2.22 ± 0.16, and 3.50 ± 0.07 μM, respectively. Compound **3** had a 28-times higher tyrosinase potency than kojic acid.

When l-dopa was used as a substrate, kojic acid exhibited an IC_50_ value of 40.42 ± 3.70 μM, which was higher compared to when l-tyrosine was used. Similar to the observation with l-tyrosine, compounds **3**, **8**, and **13,** each bearing a 2,4-dihydroxyphenyl ring, exhibited stronger tyrosinase inhibitions than kojic acid. The IC_50_ values of **3**, **8**, and **13** were 16.78 ± 0.57, 20.38 ± 1.99, and 20.76 ± 1.02 μM, respectively. The remaining compounds had IC_50_ values of >200 μΜ, except for compound **5** (IC_50_ value = 187.13 ± 30.28 μM). Among the 2-phenylbenzoxazole compounds bearing a 2,4-dihydroxyphenyl ring (**3**, **8**, and **13**), compound **3**, which belonged to series A, exerted the strongest tyrosinase inhibitory activity, regardless of the substrate type.

The structure–activity relationship is outlined in Figure 3. The presence of R^5^ played a crucial role in providing the basic tyrosinase inhibitory activity, with the hydroxyl group contributing more to tyrosinase inhibition than the methoxy group. Adding a hydroxyl group at R^4^ reduced the tyrosinase inhibitory activity. The insertion of a substituent into R^3^ induced dramatic changes in the tyrosinase inhibitory activity; the insertion of a hydroxyl group significantly increased the inhibitory potency, whereas the insertion of a methoxyl group significantly decreased the inhibitory potency. Additionally, the introduction of a methyl group at R^1^ slightly increased the inhibitory activity, more than when a methyl group was added to R^2^.

### 2.3. Inhibition Mode of Action of 2-Phenylbenzoxazole Compounds

For the inhibition mechanism study of the 2-phenylbenzoxazole compounds, kinetic studies of mushroom tyrosinase were conducted. As compounds **3**, **8**, and **13** exhibited potent inhibition activity against mushroom tyrosinase, their initial rate of dopachrome production was measured in the presence of four different compound concentrations (0, 5, 10, and 20 µM for **3** and **13** and 0, 10, 20, and 40 µM for **8**). Kinetic experiments were performed using various concentrations of l-dopa. Lineweaver–Burk plots were obtained for each compound by plotting the inverse of the l-dopa concentration against the inverse of the initial rate of dopachrome production (Figure 4). In the Lineweaver–Burk plots (Figure 4B,C), the lines for compounds **8** and **13** merged at one point on the y-axis, whereas the lines for **3** (Figure 4A) merged at one point in the second quadrant. These results suggest that **8** and **13** are competitive inhibitors whose maximum reaction rates do not change, regardless of the inhibitor concentration, and that **3** is a mixed-type inhibitor in which the maximum reaction rate decreases and the Michaelis constant increases as the inhibitor concentration increases.

For the inhibition constants (K_i_) of these compounds, each Lineweaver–Burk plot for the competitive inhibitors **8** and **13** was transformed into a corresponding Dixon plot. These Dixon plots were obtained by plotting the inhibitor concentration against the inverse of the initial dopachrome production rate (Figure 5). The four lines for each Dixon plot were merged at one point in the second quadrant. The absolute x-coordinate of each merged point represents the K_i_ value of each compound. The K_i_ values of **8** and **13** were 12.50 and 17.22 µM, respectively.

### 2.4. In Silico Docking Simulation of 2-Phenylbenoxazole Compounds **3**, **8**, and **13** and Mushroom Tyrosinase Using AutoDock Vina

Compounds **3**, **8**, and **13** showed the strongest mushroom tyrosinase inhibition. To investigate the chemical interactions and binding affinities between these 2-phenylbenoxazole compounds and mushroom tyrosinase amino acid residues, in silico docking simulations were carried out using AutoDock Vina 1.2.0.

According to the docking simulation results shown in Figure 6, compound **3** interacted hydrophobically with the amino acid residues at the tyrosinase active site. The fused benzene ring in **3** interacted with two amino acids, Ala286 and Val283, while the 6-methyl substituent interacted with three amino acids, Ala286, Val283, and Phe292. Additionally, the 2-phenyl moiety of 2-phenylbenoxazole compound **3** interacted with two amino acids, Phe264 and Val248. These hydrophobic interactions resulted in a binding affinity of −6.9 kcal/mol to compound **3**. Notably, compounds **8** and **13** also exhibited hydrophobic interactions with the same amino acid residues at the mushroom tyrosinase active site, analogous to compound **3**. The binding affinities of these compounds were −6.8 and −6.7 kcal/mol, respectively. On the other hand, kojic acid, used for comparing the binding affinity, formed hydrogen bonds with three histidine residues (His296, His61, and His263) and Met280 and interacted with His263 via pi-pi stacking, resulting in a binding affinity of −5.4 kcal/mol. The in silico docking simulation results suggest that the 2-phenylbenzoxazole compounds **3**, **8**, and **13** bound more tightly to the tyrosinase active site compared to kojic acid.

The kinetic results showed that compound **3** was a mixed-type inhibitor that could bind to both the tyrosinase active site and the allosteric site. Thus, the potential of compound **3** to bind to the tyrosinase allosteric site was assessed using a docking simulation.

Figure 7 shows compound **3**’s binding to the allosteric (red) and active (turquoise) sites. Compound **3** interacted with three amino acids (Glu67, Lys5, and Gln72) via four hydrogen bonds, and the 4-hydroxyl group of the 2-phenyl ring formed three hydrogen bonds. Notably, the 4-hydroxyl group interacted with Gln72 as a hydrogen bond acceptor and donor. These interactions provided compound **3** with a binding affinity of −7.1 kcal/mol, indicating that compound **3** could bind tightly to the tyrosinase allosteric site.

### 2.5. Cytotoxicity in B16F10 Cells

Since the 2-phenylbenzoxazole compounds **3**, **8**, and **13** were identified as being potent mushroom tyrosinase inhibitors, they were used in B16F10-cell-based experiments. Before assessing the biological impact of compounds **3**, **8**, and **13** in the B16F10 cells, the effect of these compounds on B16F10 cell viability was evaluated. The cells were treated with the compounds at concentrations of 0, 1, 2, and 5 μM for 48 and 72 h, respectively.

Compounds **3**, **8**, and **13** were not cytotoxic to the B16F10 cells after 48 h at any of the concentrations tested (Figure 8). After 72 h of treatment, compounds **8** and **13** did not show cytotoxicity at any of the concentrations tested; however, compound **3** showed weak cytotoxicity at the highest concentration tested. Despite this, the level of cytotoxicity of compound **3** at 5 μM, was minimal. Therefore, the melanin inhibition ability and cellular tyrosinase inhibitory activity of compounds **3**, **8**, and **13** were evaluated at a concentration of 5 μM.

### 2.6. Effect of 2-Phenylbenzoxazole Compounds **3**, **8**, and **13** on Melanogenesis in B16F10 Cells

We investigated whether compounds **3**, **8**, and **13**, which exhibited a strong inhibition of mushroom tyrosinase, could inhibit melanogenesis in B16F10 cells. The B16F10 cells were exposed to each compound (1, 2, and 5 μM) for 1 h. Following this, stimulators (1 μM α-MSH [α-melanocyte-stimulating hormone] and 200 μM IBMX [3-isobutyl-1-methylxanthine]) were treated for 72 h. Kojic acid (5 μM) was utilized as a comparative control.

Exposure to the stimulators greatly increased melanogenesis, but the treatment with **3**, **8**, or **13** decreased the melanin content in a concentration-dependent manner (Figure 9). At 2 μM, all compounds exerted strong anti-melanogenic effects with lower melanin content levels than kojic acid. Additionally, compounds **3** and **8** at 5 μM decreased the melanin content levels much more strongly than kojic acid at the same concentration. These results suggest that compounds **3**, **8**, and **13**, which potently inhibit mushroom tyrosinase, have the potential to inhibit melanogenesis by inhibiting cellular tyrosinase. Thus, the inhibitory effects of **3**, **8**, and **13** on B16F10 cellular tyrosinase were assessed.

### 2.7. Effect of 2-Phenylbenzoxazole Compounds **3**, **8**, and **13** on B16F10 Cellular Tyrosinase Inhibition

To investigate the origins of the anti-melanogenic effects of compounds **3**, **8**, and **13**, their impacts on B16F10 cellular tyrosinase inhibition were evaluated. The experiment utilized a similar experimental method and the same concentrations as those used in the melanin content experiment. Compounds **3**, **8**, and **13** (1, 2, and 5 μM), kojic acid (5 μM: positive control), and stimulators (1 μM α-MSH and 200 μM IBMX) were used in this experiment. Prior to treatment with the stimulators for 72 h, the test samples (**3**, **8**, **13**, and kojic acid) were pre-treated for 1 h.

The exposure of the B16F10 cells to stimulators led to a significant increase in their cellular tyrosinase activity (Figure 10). However, the addition of compounds **3**, **8**, and **13** reduced the stimulator-induced tyrosinase activity in a concentration-dependent manner. The potency of these compounds in inhibiting cellular tyrosinase activity was similar to or stronger than that of kojic acid at the same concentration. In addition, the cellular tyrosinase activity results were similar to the anti-melanogenic effects of the compounds, suggesting that the anti-melanogenic properties of these compounds are primarily related to their ability to inhibit cellular tyrosinase activity.

### 2.8. In Situ Tyrosinase Activity in B16F10 Cells

To evaluate the effect of the compounds on in situ tyrosinase activity, an experiment using B16F10 cells was performed as previously described [34,35]. Since compounds **3**, **8**, and **13** exhibited potent anti-melanogenic effects, these compounds were used in this experiment, and excess l-dopa was used as a substrate to produce melanin. B16F10 cells were treated with kojic acid, a comparative control, or compounds **3**, **8**, and **13** at 5 μM and 1, 2, and 5 μM, respectively, for 1 h. Afterward, the cells were treated with stimulators (α-MSH [1 μM] and IBMX [200 μM]) for 72 h to increase their tyrosinase activity. Excess l-dopa was added to increase their melanin production prior to the acquisition of cell images.

The stimulators led to a significant increase in melanin production (Figure 11). However, treatment with 5 μM kojic acid markedly decreased melanin formation. Compounds **3**, **8**, and **13** reduced melanin formation in a concentration-dependent manner. All compounds at 2 μM showed a stronger melanin reduction than kojic acid at 5 μM. Compared to the concentration of 5 μM, all compounds exerted a much stronger melanin production inhibitory effect than kojic acid. These results suggest that the melanin reduction effects were derived from the in situ reduction in tyrosinase activity caused by these compounds.

### 2.9. Cell Viability on HaCaT

HaCaT cells, human keratinocytes, are the primary cells comprising the human epidermis. Dermal medicines and cosmetic ingredients should not be cytotoxic to epidermal cells, including HaCaT cells. Therefore, the effects of compounds **3**, **8**, and **13** on HaCaT cell viability were investigated. HaCaT cells were exposed to each compound at concentrations of 2, 5, and 10 μM for 24 h.

None of the compounds exhibited cytotoxicity against the HaCaT cells after 24 h (Figure 12). These results suggest that these compounds are suitable for dermal applications.

### 2.10. In Vivo Depigmentation Effect of 2-Phenylbenzoxazole Compounds on Zebrafish Larvae

The 2-Phenylbenzoxazole compounds exhibited inhibitory effects on both mushroom tyrosinase enzymes and B16F10 cellular tyrosinase, leading to anti-melanogenic potency in the cells. Therefore, we investigated whether these compounds induce anti-melanogenic activity in vivo. Zebrafish embryos were used in this study. Zebrafish are very similar to humans in terms of their genetic sequence and function [36,37,38]. The zebrafish embryos were obtained through natural mating in our laboratory. The chorion of the zebrafish embryos was removed 24 h post-fertilization (hpf). After 4 h, the 2-phenylbenzoxazole compounds **8** and **13** and kojic acid, a positive control, were used. Kojic acid was used at a concentration of 20 mM and compound **13** was used at concentrations of 0.025 and 0.05 mM. In contrast, compound **8** was used at 0.025 mM, because it was judged to have a weak toxicity at 0.05 mM. Before obtaining photographs of the zebrafish larvae, the zebrafish embryos were placed in an incubator for zebrafish embryos for 48 h. Photographs were acquired from dorsal and lateral views of the zebrafish larvae.

At 76 hpf, the control zebrafish larvae displayed dark pigmentation throughout their bodies, including their eyes (Figure 13B). Exposure to kojic acid resulted in reduced pigmentation in all regions except the eyes, compared with the pigmentation in the control zebrafish larvae. Compound **13** demonstrated a concentration-dependent depigmentation effect and exhibited a significantly stronger depigmentation efficacy than kojic acid, even at a concentration (0.05 mM) 400-times lower than that of kojic acid. In particular, the pigmentation of the eyes significantly decreased. Compound **8** exhibited a strong potency at 0.025 mM, similar to **13** at 0.05 mM. According to a pigmentation area analysis using CS analyzer software version 3.0, kojic acid at 20 mM decreased the pigmentation area compared to the control (Figure 13C). Compound **13** demonstrated a similar or stronger efficacy than kojic acid at 20 mM in reducing the pigmentation area at 0.025 and 0.05 mM, respectively. Compound **8** also exhibited the smallest pigmentation area, even at a concentration of 0.025 mM, which was 800-times lower than that of kojic acid. These in vivo results suggest that the 2-phenylbenzoxazole compounds **8** and **13** are promising agents for reducing melanogenesis.

Because compound **8** showed the most potent depigmentation effect on the zebrafish larvae, the melanin quantification and tyrosinase inhibitory activity of **8** were evaluated using kojic acid as a positive control for comparison. Zebrafish embryos obtained by natural mating were dechorionated at 24 hpf and treated with compound **8** (0.025 mM) or kojic acid (20 mM) at 28 hpf. After 46 h, the melanin content and tyrosinase activity were determined.

Kojic acid significantly reduced the melanin content of the zebrafish larvae (53%) compared to that of the untreated control (100%), and compound **8** exhibited a stronger reduction in melanin content (35%) than kojic acid, even at concentrations 800-times lower than kojic acid (Figure 14B). The inhibitory effect of **8** on the tyrosinase activity in the zebrafish larvae was similar to that of melanin (Figure 14C). After treatment with kojic acid or **8**, their % tyrosinase activity were 57 and 29%, respectively, compared with the untreated control (100%), and compound **8** showed a better tyrosinase inhibitory activity than kojic acid, even at a much lower concentration. These results suggest that the depigmentation effect of **8** in zebrafish larvae can be attributed to its tyrosinase inhibitory ability.

### 2.11. Antioxidant Ability of 2-Phenylbenzoxazole Compounds **1**–**15** Using 2,2-Diphenyl-1-picrylhydrazyl (DPPH), 2,2′-Azino-bis(3-ethylbenzothiazoline-6-sulfonic Acid) (ABTS) Radical Cation, and Reactive Oxygen Species (ROS)

Since a positive relationship between their anti-melanogenic effect and antioxidant ability has been reported [39], the antioxidant efficacy of 2-phenylbenzoxazole compounds **1**–**15** was investigated.

The DPPH radical scavenging activity was assessed by mixing DPPH solution with compounds **1**–**15** and incubating them in the dark for 30 min. All test samples were evaluated at 500 μM. Except for compounds **10** and **15**, which contain a catechol group, the remaining compounds showed no or weak DPPH radical scavenging activity (Figure 15A). Compounds **10** and **15** both strongly scavenged DPPH radicals, with a scavenging activity of 86%, comparable to that of l-ascorbic acid (96%), which was used as a positive control. In contrast, despite having a catechol structure, compound **5** was found to have a weak DPPH radical scavenging effect, unlike compounds **10** and **15**.

For ABTS^+•^ scavenging experiments, ABTS^+•^ was generated by oxidizing ABTS using potassium persulfate. All test samples (Trolox [TR; positive material] and **1**–**15**) were used at a final concentration of 100 μM. Among the test samples, Trolox exhibited the highest ABTS^+•^-scavenging activity (99% inhibition) (Figure 15B). The 2-Phenylbenzoxazole compounds showed a wide range of scavenging efficacies ranging from weak to strong. Compounds **10** and **15** were the most potent ABTS^+•^ scavengers (81 and 83% inhibition, respectively). Both compounds share a common catechol structure. Interestingly, compound **5**, which has a catechol structure, was found to have a very weak ABTS^+•^ scavenging ability (13% inhibition). Similar results were obtained in the DPPH radical scavenging experiment. Compounds **3**, **8**, and **13**, which showed potent inhibitions of both mushroom and murine tyrosinases, exhibited a moderate antioxidant ability to scavenge ABTS^+•^ (57, 64, and 54% inhibition, respectively). In contrast, compounds **2**, **7**, and **12**, with no hydroxyl substituents, exhibited less than 5% inhibition. Compounds **4**, **9**, and **14,** bearing a 4-hydroxy-3-methoxyphenyl, scavenged ABTS^+•^ moderately (23–51%).

For ROS scavenging experiments, test samples (**1**–**15** and Trolox [positive control]; 40 μM) containing 3-morpholinosydnonimine (SIN-1; 10 μM) were mixed with a 2′,7′-dichlorodihydrofluorescein (DCFH) solution. After 30 min, the ROS scavenging activity of the test samples was assessed. Trolox exhibited the most potent ROS scavenging activity (Figure 15C). Among the 15 compounds tested, eight compounds, including compounds **5**, **10**, and **15**, each containing a catechol structure, demonstrated a strong to moderate ROS-scavenging activity. Notably, compound **15** showed a strong ROS scavenging efficacy comparable to that of Trolox. Interestingly, compound **5** displayed antioxidant potency with a strong ROS scavenging efficacy, in contrast to the DPPH and ABTS radical scavenging experiments.

## 3. Materials and Methods

### 3.1. Synthesis

#### 3.1.1. General Methods

Solvents and chemicals were purchased from Thermo Fisher Scientific (Waltham, MA, USA), Daejung (Siheung-si, Gyeonggi-do, Republic of Korea), and SEJIN CI Co. (Seoul, Republic of Korea). Thin-layer chromatography (TLC; Silica gel 60 F_254_) plates were purchased from Merck Millipore (Darmstadt, Germany). The NMR data of the compounds were obtained using a JEOL ECZ400S instrument (JEOL Ltd., Tokyo, Japan). The NMR peak-splitting patterns are represented as s (singlet), m (multiplet), brs (broad singlet), d (doublet), and dd (doublet of doublets).

#### 3.1.2. Preparation of 2-Phenylbenzo[d]oxazole Compounds **1**–**15**

##### General Synthetic Method for the Synthesis of **1 [40]**, **2** [41], **4**, **6**, **7**, **9**, **11** [25,42,43,44], **12** [41], and **14**

Solutions of 2-hydroxyaniline (series A: 2-hydroxy-4-methylaniline, series B: 4-chloro-2-hydroxyaniline, and series C: 2-hydroxy-5-methylaniline) and 1.0 equivalents of the appropriate benzaldehydes (4-hydroxybenzaldehyde, 2,4-dimethoxybenzaldehyde, and 3-hydroxy-4-methoxybenzaldehyde) were heated under reflux in ethyl alcohol for 14–40 h and then evaporated to yield the corresponding crude imine intermediates. Without further purification, the crude imine was treated with 1.0 equivalent 2,3-dichloro-5,6-dicyanobenzoquinone (DDQ) in dichloromethane and stirred at 19 °C for 12–16 h. After the reaction mixture was filtered through a pad of Celite^®^, the filtrate was partitioned between dichloromethane and saturated NaHCO_3_ solution, and the organic layer was evaporated. The resulting residue was purified by silica gel column chromatography using hexane/ethyl acetate (1.5:1 to 5:1) or dichloromethane/methanol (50:1 to 70:1) as the eluent to obtain **1**, **2**, **4**, **6**, **7**, **9**, **11**, **12**, and **14** in yields of 45–60%.

##### General Synthetic Method for the Synthesis of **3**, **5**, **8**, **10**, **13 [45]**, and **15** (Appendix A)

A 1.0 M BBr_3_ in dichloromethane solution (4.0 equivalent for **2**, **7**, and **12** and 3.0 equivalent for **4**, **9**, and **14**) was added dropwise to a dichloromethane solution of **2**, **7**, and **12** bearing a 2,4-dimethoxyphenyl ring or **4**, **9**, and **14** bearing a 3-hydroxy-4-methoxyphenyl ring at 0 °C. The reaction mixture was then stirred at 19 °C for 2–15 h. After, the reaction mixture was partitioned between a saturated NaHCO_3_ solution and ethyl acetate. The organic layer was evaporated, and the resultant residue was purified by silica gel column chromatography using dichloromethane/methanol (50:1 to 100:1) or hexane/ethyl acetate (1:1) as the eluent to obtain **3**, **5**, **8**, **10**, **13**, and **15** in yields of 74–95%.


*4-(6-Methylbenzo[d]oxazol-2-yl)phenol (compound **1**)*


^1^H NMR (400 MHz, DMSO-d_6_) δ 10.23 (s, 1H, OH), 7.97 (d, 2H, *J* = 8.8 Hz, 2′-H, 6′-H), 7.55 (d, 1H, *J* = 8.4 Hz, 4-H), 7.49 (d, 1H, *J* = 1.6 Hz, 7-H), 7.14 (dd, 1H, *J* = 8.4, 1.6 Hz, 5-H), 6.92 (d, 2H, *J* = 8.8 Hz, 3′-H, 5′-H), 2.41 (s, 3H, CH_3_); ^13^C NMR (100 MHz, DMSO-d_6_) δ 162.8, 161.3, 150.8, 140.1, 135.2, 129.6, 126.2, 119.2, 117.9, 116.6, 111.2, 21.8.


*2-(2,4-Dimethoxyphenyl)-6-methylbenzo[d]oxazole (compound **2**)*


^1^H NMR (400 MHz, CDCl_3_) δ 8.07 (d, 1H, *J* = 8.8 Hz, 6′-H), 7.63 (d, 1H, *J* = 8.0 Hz, 4-H), 7.34 (d, 1H, *J* = 1.6 Hz, 7-H), 7.10 (dd, 1H, *J* = 8.0, 1.6 Hz, 5-H), 6.61 (dd, 1H, *J* = 8.8, 2.4 Hz, 5′-H), 6.58 (d, 1H, *J* = 2.4 Hz, 3′-H), 4.00 (s, 3H, OCH_3_), 3.87 (s, 3H, OCH_3_), 2.47 (s, 3H, CH_3_); ^13^C NMR (100 MHz, CDCl_3_) δ 163.4, 161.2, 159.9, 150.5, 140.2, 134.9, 132.4, 125.4, 119.3, 110.5, 109.4, 105.3, 99.2, 56.3, 55.6, 21.9.


*4-(6-Methylbenzo[d]oxazol-2-yl)benzene-1,3-diol (compound **3**)*


^1^H NMR (400 MHz, DMSO-d_6_) δ 11.21 (s, 1H, OH), 10.32 (s, 1H, OH), 7.77 (d, 1H, *J* = 8.4 Hz, 6′-H), 7.59 (d, 1H, *J* = 8.0 Hz, 4-H), 7.54 (d, 1H, *J* = 1.6 Hz, 7-H), 7.18 (dd, 1H, *J* = 8.0, 1.6 Hz, 5-H), 6.47 (dd, 1H, *J* = 8.4, 2.4 Hz, 5′-H), 6.42 (d, 1H, *J* = 2.4 Hz, 3′-H), 2.42 (s, 3H, CH_3_); ^13^C NMR (100 MHz, DMSO-d_6_) δ 163.2, 162.9, 160.2, 149.2, 137.8, 135.7, 129.1, 126.6, 118.4, 111.2, 109.2, 103.3, 102.5, 21.7.


*2-Methoxy-5-(6-methylbenzo[d]oxazol-2-yl)phenol (compound **4**)*


^1^H NMR (400 MHz, CDCl_3_) δ 7.79–7.76 (m, 2H, 2′-H, 6′-H), 7.59 (d, 1H, *J* = 8.0 Hz, 4-H), 7.34 (d, 1H, *J* = 1.6 Hz, 7-H), 7.13 (dd, 1H, *J* = 8.0, 1.6 Hz, 5-H), 6.95 (d, 1H, *J* = 8.8 Hz, 5′-H), 5.79 (s, 1H, OH), 3.95 (s, 3H, OCH_3_), 2.48 (s, 3H, CH_3_); ^13^C NMR (100 MHz, CDCl_3_) δ 162.8, 151.0, 149.4, 146.0, 139.8, 135.2, 125.7, 120.6, 120.5, 119.1, 113.7, 110.8, 110.7, 56.1, 21.9.


*4-(6-Methylbenzo[d]oxazol-2-yl)benzene-1,2-diol (compound **5**)*


^1^H NMR (400 MHz, DMSO-d_6_) δ 9.68 (brs, 2H, 2 × OH), 7.53 (d, 1H, *J* = 8.0 Hz, 4-H), 7.52 (d, 1H, *J* = 2.0 Hz, 2′-H), 7.48 (d, 1H, *J* = 1.0 Hz, 7-H), 7.45 (dd, 1H, *J* = 8.4, 2.0 Hz, 6′-H), 7.12 (dd, 1H, *J* = 8.0, 1.0 Hz, 5-H), 6.87 (d, 1H, *J* = 8.4 Hz, 5′-H), 2.40 (s, 3H, CH_3_); ^13^C NMR (100 MHz, DMSO-d_6_) δ 162.9, 150.8, 149.8, 146.3, 140.1, 135.1, 126.1, 120.0, 119.1, 118.1, 116.6, 114.7, 111.1, 21.8.


*4-(6-Chlorobenzo[d]oxazol-2-yl)phenol (compound **6**)*


^1^H NMR (400 MHz, DMSO-d_6_) δ 10.36 (s, 1H, OH), 7.97 (d, 2H, *J* = 8.8 Hz, 2′-H, 6′-H), 7.87 (d, 1H, *J* = 2.0 Hz, 7-H), 7.69 (d, 1H, *J* = 8.4 Hz, 4-H), 7.36 (dd, 1H, *J* = 8.4, 2.0 Hz, 5-H), 6.93 (d, 2H, *J* = 8.8 Hz, 3′-H, 5′-H); ^13^C NMR (100 MHz, DMSO-d_6_) δ 164.2, 161.8, 150.9, 141.4, 130.0, 129.3, 125.5, 120.7, 117.2, 116.7, 111.7.


*6-Chloro-2-(2,4-dimethoxyphenyl)benzo[d]oxazole (compound **7**)*


^1^H NMR (400 MHz, CDCl_3_) δ 8.05 (d, 1H, *J* = 8.4 Hz, 6′-H), 7.65 (d, 1H, *J* = 8.4 Hz, 4-H), 7.54 (d, 1H, *J* = 2.0 Hz, 7-H), 7.27 (dd, 1H, *J* = 8.4, 2.0 Hz, 5-H), 6.61 (dd, 1H, *J* = 8.4, 2.4 Hz, 5′-H), 6.58 (d, 1H, *J* = 2.4 Hz, 3′-H), 3.98 (s, 3H, OCH_3_), 3.88 (s, 3H, OCH_3_); ^13^C NMR (100 MHz, CDCl_3_) δ 163.9, 162.4, 160.1, 150.3, 141.2, 132.6, 130.0, 124.9, 120.4, 110.9, 108.6, 105.5, 99.2, 56.3, 55.7.


*4-(6-Chlorobenzo[d]oxazol-2-yl)benzene-1,3-diol (compound **8**)*


^1^H NMR (400 MHz, DMSO-d_6_) δ 10.99 (s, 1H, OH), 10.39 (s, 1H, OH), 7.91 (d, 1H, *J* = 2.0 Hz, 7-H), 7.76 (d, 1H, *J* = 8.4 Hz, 6′-H), 7.72 (d, 1H, *J* = 8.4 Hz, 4-H), 7.40 (dd, 1H, *J* = 8.4, 2.0 Hz, 5-H), 6.48 (dd, 1H, *J* = 8.4, 2.4 Hz, 5′-H), 6.43 (d, 1H, *J* = 2.4 Hz, 3′-H); ^13^C NMR (100 MHz, DMSO-d_6_) δ 164.1, 163.6, 160.3, 149.5, 139.4, 129.7, 129.6, 125.9, 120.0, 111.8, 109.4, 103.4, 102.3.


*5-(6-Chlorobenzo[d]oxazol-2-yl)-2-methoxyphenol (compound **9**)*


^1^H NMR (400 MHz, DMSO-d_6_) δ 9.61 (s, 1H, OH), 7.86 (d, 1H, *J* = 2.0 Hz, 7-H), 7.69 (d, 1H, *J* = 8.8 Hz, 4-H), 7.57 (dd, 1H, *J* = 8.4, 2.4 Hz, 6′-H), 7.54 (d, 1H, *J* = 2.4 Hz, 2′-H), 7.36 (dd, 1H, *J* = 8.8, 2.0 Hz, 5-H), 7.07 (d, 1H, J = 8.4 Hz, 5′-H), 3.82 (s, 3H, OCH_3_); ^13^C NMR (100 MHz, DMSO-d_6_) δ 163.9, 151.8, 151.0, 147.5, 141.3, 129.5, 125.6, 120.8, 120.1, 118.8, 114.4, 112.9, 111.8, 56.2.


*4-(6-Chlorobenzo[d]oxazol-2-yl)benzene-1,2-diol (compound **10**)*


^1^H NMR (400 MHz, DMSO-d_6_) δ 7.84 (d, 1H, *J* = 2.0 Hz, 7-H), 7.65 (d, 1H, *J* = 8.4 Hz, 4-H), 7.46 (d, 1H, *J* = 2.4 Hz, 2′-H), 7.44 (dd, 1H, *J* = 8.0, 2.4 Hz, 6′-H), 7.34 (dd, 1H, *J* = 8.4, 2.0 Hz, 5-H), 6.80 (d, 1H, *J* = 8.0 Hz, 5′-H); ^13^C NMR (100 MHz, DMSO-d_6_) δ 164.7, 152.6, 150.9, 147.1, 141.6, 128.9, 125.3, 120.5, 120.4, 116.5, 115.7, 114.1, 111.6.


*4-(5-Methylbenzo[d]oxazol-2-yl)phenol (compound **11**)*


^1^H NMR (400 MHz, DMSO-d_6_) δ 10.25 (brs, 1H, OH), 7.98 (d, 2H, *J* = 8.8 Hz, 2′-H, 6′-H), 7.54 (d, 1H, *J* = 8.4 Hz, 7-H), 7.48 (d, 1H, *J* = 1.2 Hz, 4-H), 7.13 (dd, 1H, *J* = 8.4, 1.2 Hz, 6-H), 6.92 (d, 2H, *J* = 8.8 Hz, 3′-H, 5′-H), 2.38 (s, 3H, CH_3_); ^13^C NMR (100 MHz, DMSO-d_6_) δ 163.4, 161.4, 148.8, 142.5, 134.4, 129.7, 126.2, 119.7, 117.8, 116.6, 110.5, 21.5.


*2-(2,4-Dimethoxyphenyl)-5-methylbenzo[d]oxazole (compound **12**)*


^1^H NMR (400 MHz, CDCl_3_) δ 8.07 (d, 1H, *J* = 8.4 Hz, 6′-H), 7.54 (d, 1H, *J* = 2.0 Hz, 4-H), 7.40 (d, 1H, *J* = 8.4 Hz, 7-H), 7.09 (dd, 1H, *J* = 8.4, 2.0 Hz, 6-H), 6.60 (dd, 1H, *J* = 8.4, 2.0 Hz, 5′-H), 6.58 (d, 1H, *J* = 2.0 Hz, 3′-H), 3.98 (s, 3H, OCH_3_), 3.87 (s, 3H, OCH_3_), 2.45 (s, 3H, CH_3_); ^13^C NMR (100 MHz, CDCl_3_) δ 163.5, 161.9, 160.0, 148.5, 142.5, 133.9, 132.5, 125.6, 119.9, 109.6, 109.4, 105.4, 99.2, 56.2, 55.6, 21.6.


*4-(5-Methylbenzo[d]oxazol-2-yl)benzene-1,3-diol (compound **13**)*


^1^H NMR (400 MHz, DMSO-d_6_) δ 11.25 (s, 1H, OH), 10.37 (s, 1H, OH), 7.80 (d, 1H, *J* = 8.4 Hz, 6′-H), 7.59 (d, 1H, *J* = 8.4 Hz, 7-H), 7.52 (d, 1H, *J* = 1.6 Hz, 4-H), 7.17 (dd, 1H, *J* = 8.4, 1.6 Hz, 6-H), 6.47 (dd, 1H, *J* = 8.4, 2.4 Hz, 5′-H), 6.42 (d, 1H, *J* = 2.4 Hz, 3′-H), 2.39 (s, 3H, CH_3_); ^13^C NMR (100 MHz, DMSO-d_6_) δ 163.5, 163.2, 160.3, 147.2, 140.2, 135.1, 129.3, 126.5, 118.9, 110.7, 109.3, 103.4, 102.6, 21.5.


*2-Methoxy-5-(5-methylbenzo[d]oxazol-2-yl)phenol (compound **14**)*


^1^H NMR (400 MHz, DMSO-d_6_) δ 9.53 (s, 1H, OH), 7.58 (dd, 1H, *J* = 8.4, 2.0 Hz, 6′-H), 7.55 (d, 1H, *J* = 8.4 Hz, 7-H), 7.55 (d, 1H, *J* = 2.0 Hz, 4-H), 7.49 (d, 1H, *J* = 2.0 Hz, 2′-H), 7.14 (dd, 1H, *J* = 8.4, 2.0 Hz, 6-H), 7.07 (d, 1H, *J* = 8.4 Hz, 5′-H), 3.82 (s, 3H, OCH_3_), 2.38 (s, 3H, CH_3_); ^13^C NMR (100 MHz, DMSO-d_6_) δ 163.1, 151.5, 148.9, 147.4, 142.5, 134.5, 126.3, 119.8, 119.8, 119.5, 114.3, 112.9, 110.6, 56.2, 21.5.


*4-(5-Methylbenzo[d]oxazol-2-yl)benzene-1,2-diol (compound **15**)*


^1^H NMR (400 MHz, DMSO-d_6_) δ 9.72 (brs, 1H, OH), 9.45 (brs, 1H, OH), 7.53 (d, 1H, *J* = 8.4 Hz, 7-H), 7.52 (d, 1H, *J* = 2.0 Hz, 4-H), 7.48–7.45 (m, 2H, 2′-H, 6′-H), 7.11 (dd, 1H, *J* = 8.4, 2.0 Hz, 6-H), 6.87 (d, 1H, *J* = 8.0 Hz, 5′-H), 2.38 (s, 3H, CH_3_); ^13^C NMR (100 MHz, DMSO-d_6_) δ 163.5, 149.9, 148.8, 146.3, 142.6, 134.4, 126.1, 120.1, 119.6, 118.1, 116.7, 114.8, 110.5, 21.5.

### 3.2. Reagents for Biological Experiments

Potassium hydrogen phosphate, potassium dihydrogen phosphate, phenylmethylsulfonyl fluoride, 3-morpholinosydnomimine (SIN-1), 2,2-azino-bis(3-ethylbenzothiazoline-6-sulfonic acid) (ABTS), α-melanocyte-stimulating hormone (α-MSH), mushroom tyrosinase, dimethyl sulfoxide (DMSO), l-tyrosine, and l-dopa were purchased from Sigma-Aldrich (St. Louis, MO, USA).

### 3.3. Tyrosinase Activity Assay Using Mushroom Tyrosinase

The tyrosinase inhibitory activity of the 2-phenylbenzoxazoles was evaluated using a mushroom tyrosinase activity assay, following a previously described method [46]. An aqueous substrate mixture, consisting of phosphate buffer (pH 6.5; 17.2 mM) and l-tyrosine or l-dopa (345 μM), was prepared. A 96-well plate containing the test sample in dimethyl sulfoxide (DMSO) solution (10 μL), aqueous mushroom tyrosinase solution (20 μL; 200 units), and aqueous substrate mixture (170 μL) in each well was incubated at 37 °C for 15 min for l-dopa and 30 min for l-tyrosine, respectively. Using a VersaMax^®^ ELISA reader (VersaMax Pro 6.5.1; Molecular Devices, Sunnyvale, CA, USA), the well absorbance at 475 nm was recorded to calculate the % tyrosinase activity inhibition: % tyrosinase activity inhibition = (1 − Abs_sam_/Abs_con_) × 100, where Abs_sam_ and Abs_con_ are the absorbances of the samples and control, respectively. Kojic acid was used to compare tyrosinase inhibitory activity. For IC_50_ values, test samples were used at three concentrations (4, 20, and 100 μM for **1**, **2**, **4**, **5**, **7**, and **9**–**12**, 2, 10, and 50 μM for **6**, **14**, **15**, and kojic acid, 0.16, 0.8, and 4 μM for **8** and **13**, and 0.032, 0.8, and 4 μM for **3** in the presence of l-tyrosine; 4, 20, and 100 μM for **1**, **2**, **4**–**7**, **9**–**12**, **14**, **15**, and kojic acid and 0.8, 4, and 20 μM for **3**, **8**, and **13** in the presence of l-dopa).

### 3.4. Kinetic Study Experiment Using Mushroom Tyrosinase

Mushroom tyrosinase was incubated with 2-phenylbenzoxazole inhibitors (**3**, **8**, or **13**) in the presence of various concentrations (0.5–16 mM) of l-dopa to measure the initial dopachrome production rate. An aqueous substrate mixture, consisting of phosphate buffer (pH 6.5; 17.2 mM) and l-dopa (345 μM), was prepared. A 96-well plate containing the inhibitor DMSO solution (10 μL), aqueous mushroom tyrosinase solution (20 μL; 20 units), and aqueous substrate mixture (170 μL) in each well was incubated for 20 min at 37 °C. During incubation, the absorbance at 475 nm was recorded at 5 min intervals using an ELISA reader to prepare Lineweaver–Burk plots for each inhibitor.

### 3.5. Docking Simulation Using AutoDock Vina

The 2-phenylbenzoxazole compounds **3**, **8**, and **13**, which exhibited the most potent mushroom tyrosinase inhibitory activities, were selected for the docking simulation. The ligand 3D structure for the docking simulation was prepared using Chem3D Pro 12.0. The 3D X-ray crystal structure of mushroom tyrosinase (PDB ID: 2Y9X) was obtained from the RCSB Protein Data Bank (PDB). Docking simulations between the ligand (2-phenylbenzoxazoles **3**, **8**, and **13** and kojic acid [positive material]) and tyrosinase were conducted using AutoDock Vina 1.2.0 after the original ligand, tropolone, was removed from the active site. Chimera 1.13.1 and LigandScout 4.3 were utilized to obtain information on the plausible chemical interactions between the ligand and tyrosinase amino acid residues.

### 3.6. Cell Culture

B16F10 cells and HaCaT cells, obtained from the American Type Culture Collection (VA, USA), were cultured under an environment including 5% CO_2_ at 37 °C in Dulbecco’s modified Eagle’s medium, including 10% heat-inactivated FBS and penicillin–streptomycin (100X) solution (Welgene, Gyeongsangbuk-do, Republic of Korea).

### 3.7. B16F10 Cell Viability Assay

The influence of compounds **3**, **8**, and **13** on cytotoxicity was measured in B16F10 cells using a previously described method [47]. The B16F10 cells were inoculated at a density of 1 × 10^3^ cells per well and cultured for 24 h in a 96-well plate under the same incubation conditions used for the cell culture. Test samples (**3**, **8**, and **13**) were added to each well at final concentrations of 0, 1, 2, and 5 μM and cultivated for 48 and 72 h, respectively. Each well was treated with 10 μL of EZ-Cytox solution (EZ-1000^®^, DoGenBio, Seoul, Republic of Korea) and cultivated for 2 h. The absorbance of each well was measured at 450 nm using an ELISA reader.

### 3.8. Cellular Melanin Content Level Assay in B16F10 Cells

The effects of compounds **3**, **8**, and **13** on B16F10 cellular melanin production were investigated using a previously reported method [48]. B16F10 cells (5 × 10^3^ cells per well) were seeded in a 6-well plate and cultured under the same conditions as the cell culture for 24 h. The cells were pre-treated with test samples (kojic acid [positive control]: 5 μM or compounds **3**, **8**, and **13**: 1, 2, and 5 μM) for 1 h. Stimulators, including 1 μM α-melanocyte-stimulating hormone (α-MSH) and 200 μM 3-isobutyl-1-methylxanthine (IBMX), were then added. After 72 h of cultivation, the cells were rinsed with PBS, detached from the bottom of each well using Trypsin-EDTA, and centrifuged for 10 min at 10,000× *g* and 4 °C. The obtained pellets were lysed with 100 μL of 1N-NaOH solution for 1 h at 60 °C and transferred to a 96-well plate. The optical density was measured at 405 nm using an ELISA reader. Normalization was conducted using a Pierce BCA Protein Assay Kit (Thermo Scientific, MA, USA).

Melanin content (%) = (ΔOD_sam_/ΔOD_con_) × 100%, where OD_sam_ and OD_con_ represent the optical densities of the sample and control, respectively.

### 3.9. Cellular Tyrosinase Activity Assay in B16F10 Cells

The effects of compounds **3**, **8**, and **13** on B16F10 cellular tyrosinase activity were assessed using a previously reported method [48]. B16F10 cells were seeded, treated with the samples and stimulators, and cultured for 72 h in the same manner as for the cellular melanin content assay. To lyse the cultured cells, a lysis buffer was prepared containing 1 mM phenylmethylsulfonyl fluoride, 50 mM phosphate buffer, and 1% Triton X-100 (5:90:5, *v*/*v*/*v*). The cultivated cells were lysed by adding the prepared lysis buffer (100 µL) after washing twice with PBS. The lysed cells were centrifuged for 30 min at 4 °C and 10,000 *g* after freezing for 1 h at −80 °C. The supernatants (80 µL) and 10 mM l-dopa (20 µL) were added to each well of a 96-well plate and cultivated for 10 min at 37 °C. The optical density was measured at 475 nm using an ELISA reader. Normalization was performed using a Pierce BCA Protein Assay Kit.

### 3.10. In Situ Tyrosinase Activity Assay Using B16F10 Cells and l-dopa

In situ tyrosinase activity was measured in B16F10 cells using l-dopa, as reported previously [34]. Compounds **3**, **8**, and **13**, which exhibited potent melanin inhibition in the B16F10 cells, were tested at 1, 2, and 5 μM. The B16F10 cells were seeded at a density of 1 × 10^3^ cells per well in a 24-well microplate at 37 °C with 5% CO_2_ for 24 h. Before exposure to stimulators (200 μM IBMX and 1 μM α-MSH), the cells were pre-treated with test samples (compounds **3**, **8**, and **13** or kojic acid [positive material; 5 μM]) for 1 h. The cells were processed as follows: (1) fixing with 4% paraformaldehyde for 40 min, (2) washing with PBS, and (3) permeabilizing with 0.1% Triton X-100 for 2 min. After washing with PBS, the cells were stained with l-dopa (2 mM; 500 μL) at 37 °C for 2 h. Stained images were collected using a camera connected to a microscope (Motic, Hong Kong).

### 3.11. HaCaT Cell Viability Assay

The cytotoxicity of compounds **3**, **8**, and **13** was assessed in HaCaT cells using a previously described method [38]. The HaCaT cells were inoculated at a density of 1 × 10^4^ cells/well in a 96-well plate and cultured for 24 h under the standard incubation conditions. Test samples (**3**, **8**, and **13**) were added at the final concentrations of 0, 2, 5, and 10 μM and incubated for an additional 24 h. Each well was treated with 10 μL of EZ-Cytox solution and cultivated for 2 h. The absorbance of each well was measured at 450 nm using an ELISA reader.

### 3.12. In Vivo Depigmentation Experiment Using Zebrafish Embryos

A depigmentation experiment was performed using zebrafish embryos, as previously described [38,49,50]. Wild-type zebrafish (*Danio rerio*) were maintained in a fish tank at 28 °C and fed with dried brine shrimp (*Artemia salina*, San Francisco Bay Brand, San Francisco, CA, USA) three times daily. The zebrafish embryos were obtained through natural mating in mating cages. The obtained zebrafish embryos were transferred to a 90 mm culture dish containing 200 μL of E3-methylene blue (MB) solution, consisting of 0.001% MB, 0.17 mM KCl, 0.33 mM CaCl_2_, 0.33 mM MgSO_4_, and 5 mM NaCl, and incubated in an incubator set at 28 °C. At 24 h post-fertilization (hpf), the zebrafish embryos were dechorionated using pronase (Sigma-Aldrich, St. Louis, MO, USA). Test samples (20 mM kojic acid, a positive control, 0.025 mM compound **8**, and 0.025 and 0.05 mM compound **13**) were added to each well of a 48-well plate containing four dechorionated zebrafish embryos and 200 μL of E3 solution per well. The 48-well plates were incubated in an incubator set at 28 °C for 48 h. At 76 hpf, the zebrafish larvae were mounted on a 1% methylcellulose block after anesthesia with tricaine methanesulfonate (Thermo Fisher Scientific, Waltham, MA, USA), and photographs of the zebrafish larvae were obtained using a SMZ745T stereoscopic microscope (Nikon, Tokyo, Japan). The pigmented areas of the zebrafish larvae were determined using the CS analyzer 3.0 image analysis software (ATTO, Tokyo, Japan).

### 3.13. Measurement of Melanin Content and Tyrosinase Activity in Zebrafish Larvae

The effects of the 2-phenylbenzo[*d*]oxazole compound **8** on the melanin content and tyrosinase activity in the zebrafish larvae were assessed using a previously defined procedure with minor modifications [49,51]. To quantify the melanin content in the zebrafish larvae, zebrafish embryos were obtained through natural mating and operated in a similar manner to the in vivo depigmentation experiment (25 zebrafish embryos per well in a 24-well plate; dechorionation at 24 hpf; test sample treatment (**8** [0.025 mM] and kojic acid [20 mM; positive control]) at 28 hpf). Melanin quantification and tyrosinase activity were quantified at 74 hpf. For melanin quantification, the zebrafish larvae were lysed in RIPA buffer (Biosesang, Gyeonggi-do, Korea) and homogenized using a sonicator. After centrifuging the lysates (10,000× *g* and 10 min), the pellet was dissolved in 100 μL of 1N NaOH at 100 °C for 30 min and roughly vortexed to solubilize the melanin. The absorbance was measured at 405 nm using a microplate reader. For the tyrosinase activity measurement, the supernatants were transferred to new Eppendorf tubes and the protein was quantified using a Pierce BCA protein assay kit, with the absorbance being measured at 570 nm using a microplate reader. A total of 500 μg of protein in 100 μL of protein lysate was transferred into each well of a 96-well plate, followed by 100 μL of 1 mM l-dopa. The mixture was incubated for 1 h at 37 °C. The absorbance was measured at 475 nm to determine the tyrosinase activity. Kojic acid was used as a positive control.

### 3.14. 2,2′-Diphenyl-1-picrylhydrazyl (DPPH) Radical Scavenging Assay

As previously reported [52,53], the ability of compounds **1**–**15** to remove DPPH radicals was evaluated. A 0.2 mM methanolic DPPH solution was mixed with a 5 mM l-ascorbic acid aqueous solution or 5 mM DMSO solution of **1**–**15** at a ratio of 10:1 (*v*/*v*, 180 μL:20 μL) in each well of a 96-well microplate. After incubating the microplate in the dark for 30 min, the optical density of each well was measured at 517 nm using an ELISA reader.
DPPH scavenging activity (%) = (OD_[control]_ − OD_[sample]_) × 100/OD_[control]_.

### 3.15. 2,2′-Azino-bis(3-ethylbenzothiazoline-6-sulfonic Acid) (ABTS) Radical Cation Scavenging Assay

The effect of the 2-phenylbenzoxazole compounds **1**–**15** on radical scavenging was assessed using an ABTS assay, as previously reported [54]. An ABTS aqueous solution (7 mM) was mixed with a K_2_S_2_O_8_ aqueous solution (2.45 mM) in equal volumes (each 20 mL) and incubated in the dark for 20 h at 25 °C to generate ABTS^+•^. The ABTS^+•^ solution was then diluted with pure EtOH to adjust the absorbance of the solution to 0.7 ± 0.02 at 730 nm. The test samples (**1**–**15** and Trolox [positive control]) were dissolved in a co-solvent of EtOH and DMSO (9:1, *v*/*v*) to make a 1 mM solution. The diluted ABTS^+•^ solution was mixed with the test sample solution at a volume ratio of 9:1 (90 μL:10 μL) and the mixture was stored for 2 min at 20 °C in the dark. The optical density was measured at 730 nm using an ELISA reader at 1 min intervals for 10 min. All test samples were used at a final concentration of 100 µM.

ABTS^+•^ radical scavenging activity (%) = [(OD_con_ − OD_sam_)/OD_con_] × 100, where OD_sam_ and OD_con_ represent the absorbances of the test sample and the control, respectively.

### 3.16. Reactive Oxygen Species (ROS) Scavenging Assay

The ability of compounds **1**–**15** to remove ROS was assayed as previously reported [55]. 2′,7′-Dichlorodihydrofluorescein diacetate (DCFH-DA; 1.25 mM, 50 μL) was treated with esterase (0.6 units/μL, 50 μL) in phosphate pH 7.4 buffer (50 mM, 4.9 mL) and left at 25 °C for 30 min to generate DCFH. A 3-morpholinosydnonimine (SIN-1) solution (10 μL, 250 μM) and phosphate buffer (180 μL) were mixed with a test sample (**1**–**15**; 10 μL, 1 mM) in DMSO solution or Trolox (10 μL, 1 mM; positive control) in each well of a 96-well black plate and left in the dark 25 °C for 5 min. The SIN-1-test sample mixture was then combined with the generated DCFH solution (50 μL). Fluorescence was measured in 5 min intervals for 30 min at 530 nm using a microplate reader (Berthold Advances GmbH & Co., Bad Wildbad, Germany) with excitation at 485 nm.

### 3.17. Statistical Analysis

Results from three independent experiments are presented as the mean ± SEM. The statistical significance between groups was assessed using one-way ANOVA followed by the Newman–Keuls test using GraphPad Prism 5 (La Jolla, CA, USA). *p* < 0.05 was considered to be significant.

## 4. Conclusions

Compounds with a 2-phenylbenzo[*d*]thiazole scaffold exhibit potent tyrosinase inhibitory activity against murine and mushroom tyrosinases, and 2-phenylbenzo[*d*]thiazole is bioisosteric with 2-phenylbenzo[*d*]oxazole. Therefore, we designed and synthesized 15 compounds with a 2-phenylbenzo[*d*]oxazole scaffold as potential tyrosinase inhibitors. Among them, compounds **3**, **8**, and **13** demonstrated potent mushroom tyrosinase inhibition, with compound **3** having a nanomolar IC_50_ value of 0.51 ± 0.00 μM, which was significantly lower than that of kojic acid (IC_50_ value: 14.33 ± 1.63 μM). Kinetic studies confirmed that **3** is a mixed-type inhibitor, while **8** and **13** are competitive inhibitors. Docking simulations identified the chemical interactions occurring between the amino acid residues of tyrosinase and these compounds. In B16F10-cell-based experiments, compounds **3**, **8**, and **13** inhibited cellular tyrosinase activity and melanin production more effectively than kojic acid. Additionally, a staining method using a l-dopa to measure in situ tyrosinase activity also demonstrated that these compounds had stronger B16F10 cellular tyrosinase inhibitory activity than kojic acid. Compounds **3**, **8**, and **13** also exhibited superior depigmentation effects on zebrafish larvae, even at concentrations hundreds of times lower than those of kojic acid. Although the 2-phenylbenzo[*d*]oxazole derivatives were shown to be promising tyrosinase inhibitors, there are structural differences between the tetramer of mushroom tyrosinase and the glycosylated monomer of human tyrosinase. In addition, pharmacokinetic parameters cannot be predicted using simple animal models. Therefore, further studies are needed for clinical application.

## Data Availability

The data presented in this study are available in article and Appendix A.

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
