# Peer review of "Exploration of Compounds with 2-Phenylbenzo[d]oxazole Scaffold as Potential Skin-Lightening Agents through Inhibition of Melanin Biosynthesis and Tyrosinase Activity"

_molecules, 2024, doi:10.3390/molecules29174162_

Round 1

Reviewer 1 Report

Comments and Suggestions for Authors

I consider the organic synthesis part of the work and the physical-chemical characterization of the compounds correct.

Compounds 3, 8 and 13 must be numbered and this numbering will be reflected in the NMR spectra (line 487, line 515 and line 544).

Comparison between IC50 values ​​obtained using the equation

IC50= (1+n)KI                                                                                                                                            1

And the experimental values ​​for compounds 8 and 13     

. The KI values ​​are 12.50 µM and 17.22 µM (line 198). The IC50 (µM) values ​​described in Table 1 for diphenolase activity are: 20.38± 1.99 and 20.76±1.02 respectively. These values ​​satisfy the relationship [1]:

IC50= (1+n)KI 1   

Taking into account that “n” is equal to [L-dopa concentration] / Michaelis constant for L-dopa.

   IC50= (1+n)KI= (1+n)12.50=(1+0.58)12.50=19.75 µM for compound 8

   IC50= (1+n)KI= (1+n)17.22=(1+0.58)17.22=27.21 µM for compound 13.

As the experimental IC50 values ​​for diphenolase activity are 20.38± 1.99 µM and 20.76±1.02 µM, they approximately coincide with the values ​​calculated according to Equation 1.

In the case of monophenolase activity, the results do not coincide, therefore we recommend either removing them or doing them with a correct experimental design, that is, adding L-dopa at the beginning of the reaction [2].

References

[1] Pablo García Molina et al.  The Relationship between the IC50 Values and the Apparent Inhibition Constant in the Study of Inhibitors of Tyrosinase Diphenolase Activity Helps Confirm the Mechanism of Inhibition. Molecules  2022 May 13;27(10):3141.

[2] Pablo García Molina et al. Considerations about the inhibition of monophenolase and diphenolase activities of tyrosinase. Characterization of the inhibitor concentration which generates 50 % of inhibition, type and inhibition constants. A review Int J Biol Macromol . 2024 May;267(Pt 2):131513

Author Response

Comment 1: I consider the organic synthesis part of the work and the physical-chemical characterization of the compounds correct.

Compounds 3, 8 and 13 must be numbered and this numbering will be reflected in the NMR spectra (line 487, line 515 and line 544).

Response 1: Thank you for your valuable comment.

When numbering compounds by their IUPAC names, different numbers are assigned to compounds even at the same position on the phenyl ring. It is easier for readers to understand if the numbering is done so that the same positions on the phenyl ring are assigned the same numbers, excluding the compound names in the NMR portion. Therefore, we have added in the text that the numbering of the phenyl ring at position 2 of benzoxazole followed the numbering of the common phenyl ring.

Comment 2: Comparison between IC50 values â€‹â€‹obtained using the equation

IC50= (1+n)KI                                                                                                                                            1

And the experimental values â€‹â€‹for compounds 8 and 13     

. The KI values â€‹â€‹are 12.50 µM and 17.22 µM (line 198). The IC50 (µM) values â€‹â€‹described in Table 1 for diphenolase activity are: 20.38± 1.99 and 20.76±1.02 respectively. These values â€‹â€‹satisfy the relationship [1]:

IC50= (1+n)KI 1   

Taking into account that “n” is equal to [L-dopa concentration] / Michaelis constant for L-dopa.

   IC50= (1+n)KI= (1+n)12.50=(1+0.58)12.50=19.75 µM for compound 8

   IC50= (1+n)KI= (1+n)17.22=(1+0.58)17.22=27.21 µM for compound 13.

As the experimental IC50 values â€‹â€‹for diphenolase activity are 20.38± 1.99 µM and 20.76±1.02 µM, they approximately coincide with the values â€‹â€‹calculated according to Equation 1.

In the case of monophenolase activity, the results do not coincide, therefore we recommend either removing them or doing them with a correct experimental design, that is, adding L-dopa at the beginning of the reaction [2].

References

[1] Pablo García Molina et al.  The Relationship between the IC50 Values and the Apparent Inhibition Constant in the Study of Inhibitors of Tyrosinase Diphenolase Activity Helps Confirm the Mechanism of Inhibition. Molecules  2022 May 13;27(10):3141.

[2] Pablo García Molina et al. Considerations about the inhibition of monophenolase and diphenolase activities of tyrosinase. Characterization of the inhibitor concentration which generates 50 % of inhibition, type and inhibition constants. A review Int J Biol Macromol . 2024 May;267(Pt 2):131513

Response 2: Thank you for your kind comments and suggestion.

Unlike diphenolase, in the monophenolase experiments a lag period is needed to reach the steady state. So, we acquired absorbance of each well after 30 min incubation corresponding to the lag period. The reviewer recommended adding L-dopa at the beginning of the reaction. However, since we don’t have any information on how much L-dopa should be added, the experiment couldn’t be repeated. After performing several experiments to obtain data on the amount of L-dopa to be added, we would be able to obtain accurate results by using that method. Thus, we have written the following under Table 1 as a current alternative: When L-tyrosine was used as a substrate, the test sample, substrate solution, and tyrosinase were mixed, and the optical density was measured at 470 nm after 30 min to consume the lag time. Please understand that we could not repeat the experiment because we did not know the amount of L-dopa to add. We have also added the reference [2] above along with the above sentence.

Reviewer 2 Report

Comments and Suggestions for Authors

The study evaluated the activity of 2-phenylbenzo[d]oxazole derivatives in both in vitro and in vivo models for their whitening properties. This research is significant because current inhibitors often have side effects, such as the carcinogenic risks associated with hydroquinone and arbutin, and the allergenic reactions caused by azelaic acid and kojic acid. 2-phenylbenzo[d]oxazole derivatives could potentially be used in cosmetology, medicine, pharmacy, and the food industry.

Introduction

55-57 add citations

Materials and Methods

- The reagents used in the study should be better described.

- Antioxidant methods (volumes, concentrations, etc.) should be described in more detail.

- Add the concentration of compounds used in the anti-tyrosinase test.

- The formulas should be removed from the text and added below the text along with a description of the abbreviations.

- In situ tyrosinase activity assay using B16F10 cells and L-dopa – Why did you used only L-DOPA? The use of different substrates (tyrosine, L-DOPA) affects the result

Conclusion

The results suggest the potent activity of 2-phenylbenzo[d]oxazole derivatives as novel tyrosinase inhibitor candidates. However, the authors should add more information about the limitations of their study, including that the structure of human tyrosinase is different from that of mushroom tyrosinase, using a simple animal model, pharmacokinetic parameters cannot be predicted. In sum, further research is needed

Author Response

Comment 1:

The study evaluated the activity of 2-phenylbenzo[d]oxazole derivatives in both in vitro and in vivo models for their whitening properties. This research is significant because current inhibitors often have side effects, such as the carcinogenic risks associated with hydroquinone and arbutin, and the allergenic reactions caused by azelaic acid and kojic acid. 2-phenylbenzo[d]oxazole derivatives could potentially be used in cosmetology, medicine, pharmacy, and the food industry.

Introduction

55-57 add citations

Response 1:

Thank you for your suggestion. We have added four references corresponding to lines 55-57.

[Mutat. Res. Genet. Toxicol. Environ. Mutagen 2015, 780, 111-116; J. Cosmet. Dermatol. 2005, 4, 55-59; J. Med. Chem. 2018, 17, 7395-7418; Chem. Biol. Interact. 2005, 153, 267-270]

Comment 2:

Materials and Methods

- The reagents used in the study should be better described.

Response 2:

Thank you for your kind comment. According to the reviewer’s comment, we have added section “3.2. Reagents for biological experiments” in Materials and Methods.

Comment 3:

- Antioxidant methods (volumes, concentrations, etc.) should be described in more detail.

Response 3:

Thank you for your comment. As suggested by the reviewer’s comment, specific information on volumes, concentrations, etc. is described in the antioxidant methods. Please check out the antioxidant methods section.

Comment 4:

- Add the concentration of compounds used in the anti-tyrosinase test.

Response 4:

Thank you for your valuable comment. As suggested by the reviewer’s comment, the concentrations of compounds used were added in the anti-tyrosinase test.

Comment 5:

- The formulas should be removed from the text and added below the text along with a description of the abbreviations.

Response 5:

Thank you for your kind suggestion. As suggested by the reviewer’s suggestion, the formulas were added below the text along with a description of the abbreviations.

Comment 6:

- In situ tyrosinase activity assay using B16F10 cells and L-dopa – Why did you used only L-DOPA? The use of different substrates (tyrosine, L-DOPA) affects the result

Response 6:

Thank you for your keen inquiry.

We have never seen a paper that used L-tyrosine instead of L-dopa when measuring cellular tyrosinase activity using B16F10 cells. However, we were curious and tried measuring cellular tyrosinase activity using L-tyrosine. Unlike when L-dopa was used, when L-tyrosine was used as a substrate, the cellular tyrosinase activity was not inhibited regardless of the inhibitor concentration. Although it was only a one-time experiment, at that time we thought that this might be related to the lag (delay) time when we used L-tyrosine as a substrate. Therefore, in the in situ tyrosinase inhibitory activity experiment, we performed the experiment using L-dopa as a substrate, as used by other researchers.

Comment 7:

Conclusion

The results suggest the potent activity of 2-phenylbenzo[d]oxazole derivatives as novel tyrosinase inhibitor candidates. However, the authors should add more information about the limitations of their study, including that the structure of human tyrosinase is different from that of mushroom tyrosinase, using a simple animal model, pharmacokinetic parameters cannot be predicted. In sum, further research is needed

Response 7:

Thank you for your kind comments.

As suggested by the reviewer’s comments, the following sentences have been added to Conclusion:

Although 2-phenylbenzo[d]oxazole derivatives were shown to be promising tyrosinase inhibitors, there are structural differences between the tetramer of mushroom tyrosinase and the glycosylated monomer of human tyrosinase. In addition, pharmacokinetic parameters cannot be predicted using simple animal models. Therefore, further studies are needed for clinical application.

Round 2

Reviewer 1 Report

Comments and Suggestions for Authors

.

Reviewer 2 Report

Comments and Suggestions for Authors

Accept